# Development of a Remote Health Monitoring System to Prevent Frailty in Elderly Home-Care Patients with COPD

**DOI:** 10.3390/s22072670

**Published:** 2022-03-30

**Authors:** Chisato Ohashi, Shunsuke Akiguchi, Mineko Ohira

**Affiliations:** 1Department of General Education, National Institute of Technology, Toyama College, Toyama 939-0293, Japan; c-ohashi@nc-toyama.ac.jp; 2Department of Electronics and Computer Engineering, National Institute of Technology, Toyama College, Toyama 939-0293, Japan; 3Internal Medicine, National Hospital Organization, East Nagano Hospital, Nagano 381-8567, Japan; sp975dq9@forest.ocn.ne.jp

**Keywords:** COPD, frailty, home-care patients, health monitoring, physical activity

## Abstract

Chronic obstructive pulmonary disease (COPD) is the general term used to describe respiratory diseases such as chronic bronchitis or emphysema of the lungs. COPD is known to cause the onset of frailty due to limitations of physical activity (PA) in daily life and undernutrition. Here, we report the development process of a remote health monitoring and support system employing a tablet computer (iPad), that was created to help prevent frailty in elderly home-care patients with COPD, and the results of its use by two elderly home-care COPD patients. There was a significant increase in PA duration in one participant after use of the system, compared to before use (15.2 min (8.9) vs. 24.2 min (7.4), *p* < 0.001). PA duration also increased in the other participant (39.7 (8.1) vs. 42.9 (12.9) min; 8.1%), although the difference was not statistically significant. The system enabled recognition of patients’ behavior modifications to promote health. It is difficult to obtain quantitative data for health support, such as for respiratory rehabilitation in elderly COPD patients living at home. However, the present results suggest that virtually connecting patients with their support networks via information and communication technology (ICT) equipment provides support for the physical aspect of their care.

## 1. Introduction

Respiratory diseases are a common cause of physical limitations in daily life. They can lead to a vicious circle of decline in quality of life (QOL), which further promotes disease progression [1]. Chronic obstructive pulmonary disease (COPD) is the general term used to describe respiratory diseases, such as chronic bronchitis or emphysema of the lungs, resulting from a previous smoking habit [2]. COPD is known to cause a decline in QOL and the onset of frailty due to limitations of physical activity (PA) and undernutrition.

The Japanese Ministry of Health has stated the importance of countermeasures against COPD for extending healthy life expectancy. The government of Japan declared the “Healthy Japan 21” program in 2000 and set target values for various health indices. The second Healthy Japan 21 program, formulated in 2012, added COPD as a major lifestyle-related disease, along with cancer, cardiovascular diseases and diabetes. The government also set a target value for COPD awareness, raising it to >80% by 2022 [3]. However, the Global Initiative for Chronic Obstructive Lung Disease (GOLD) reported that COPD awareness in Japan remained at 28% as of December 2020 [4], which indicates that large numbers of Japanese people with subjective symptoms such as breathlessness, cough, or phlegm have not noticed that they have symptoms of COPD. Therefore, it is likely that initial diagnostic testing and subsequent treatment are delayed in Japanese people, which increases the risk of severe disease.

The following targets have been identified in the management of COPD: (1) improvement of the patients’ symptoms and QOL; (2) maintenance and improvement of exercise and PA capacity; (3) preventing acute exacerbations; (4) halting progression of COPD; (5) preserving and treating pulmonary complications; and (6) improving life prognosis. Breathing exercises, exercise therapy and nutritional care are the central non-pharmacotherapeutic interventions for patients with COPD. Additionally, home oxygen therapy (HOT) is used in cases with progressive hypoxemia due to chronic respiratory failure, to reduce symptoms, improve QOL and improve prognosis [5].

Once present, frailty leads to a weakening of mental and physical vitality, mobility and cognitive function, and is considered a comorbidity for other chronic diseases. Vital function deteriorates and fragility of the mind and body occur in the intermediate state between being healthy and requiring nursing care. However, it is possible to reverse frailty to some degree by adopting interventions and early treatment. Frailty has the following five standard symptoms: (1) weight loss; (2) susceptibility to fatigue; (3) reduction in walking speed; (4) grip strength weakness; and (5) reduction in physical activity. The presence of three or more of these symptoms confers a diagnosis of frailty, whereas the presence of one or two of these indicates pre-frailty [6].

A previous study reported very high prevalence rates of pre-frailty and frailty, of 56% and 20%, respectively, in elderly persons with COPD. One of the reasons for this situation is that dyspnea, a symptom of COPD, leads to a state of chronic undernutrition and a decline in PA. A combination of exercise and nutrition therapy has been reported as being effective in preventing frailty in patients with COPD [7]. Exercise therapy is known to reduce respiratory distress and depression, and to improve exercise tolerability, respiratory strength, activities of daily living (ADL) and QOL in patients with COPD [8]. The positive effects of home health care exercise as a part of respiratory rehabilitation have been verified in patients with COPD [9]. However, the rate of exercise therapy implementation among elderly patients is low [8]. One of the reasons for this is thought to be decreased motivation to participate in exercise therapy because of feelings of alienation from society or the community.

However, recent research has reported that public participation has a strong positive effect on preventing frailty in healthy elderly persons [10]. Another study found that in-home elderly persons who participate in nearby community activities maintain higher rates of mental and social health and have lower rates of decline in high-level vital functions [11]. Accordingly, we consider that the following three points are vitally important in health support that aims to prevent frailty in elderly patients with COPD: (1) exercise; (2) nutrition; and (3) public participation [10]. Studies on telemedicine for home-care patients with COPD are currently underway. Remote systems, such as monitoring of physiological parameters, including respiratory rate, blood pressure and blood oxygen saturation [12], or monitoring of the physical condition using interview items [13] etc., have already been developed. However, it is necessary to consider the issues of whether these devices are easily operable and inexpensive for patients, since most COPD patients are middle-aged and elderly. In addition, it is important to support the maintenance of exercise motivation, which is very effective in patients living at home with COPD. We developed a digital system to virtually support improvement and maintenance of the physical activity levels of elderly COPD patients. Our goals in the development of this health support system were to:-Devise a system that is inexpensive and easily operable by the elderly.-Provide more opportunities for elderly patients who have difficulty going out to communicate with people by using the system.-Provide a platform that will enable support staff to not only manage the patient’s health care, but also provide health education.-Facilitate support staff’s understanding of the daily medication of elderly home-care patients.-Present the patients’ data in a visual form, to facilitate understanding of their own physical condition by the patients.-Create a record that can be jointly reviewed by the patient and doctor at the patient’s monthly examinations, to enable the determination of future treatment strategies.

## 2. Purpose

Since the rate of frailty in elderly patients with COPD is very high [7], which in turn worsens the symptoms of COPD, it is very important to both prevent frailty and manage patients’ symptoms and body condition. To prevent frailty and deterioration of symptoms among elderly patients with COPD on a long-term basis, the existing comprehensive health support system enables family doctors and visiting nurses to share a patient’s data and to follow up with patients in a medical setting. In the case of an acute exacerbation of COPD, there can be a striking deterioration in symptoms, such as breathlessness, respiratory distress and cough. Therefore, it is desirable to monitor the condition of home-care patients with COPD on a daily basis, to enable early detection of any deterioration in symptoms and prevent an acute exacerbation.

For these reasons, we developed a remote health monitoring and support system that uses information and communication technology (ICT), with the aim of assessment and management of the physical condition and PA levels of home-care patients with COPD.

## 3. Development of the Health Condition Monitoring and Support System

We developed a remote system for monitoring the health condition and providing home health care support for elderly patients with COPD. The aim was to improve their PA habits and to prevent their condition from worsening, which can lead to other health problems, such as frailty and depression [11]. The system is outlined in Figure 1. We decided to utilize a tablet computer as the system device and, accordingly developed an application to handle input and transfer of the following data: six assessment items regarding extent of symptoms (cough, phlegm, breathing, sleep, appetite, vitality), daily step count, and energy expenditure. The six symptom assessment items were developed by a doctor and nurses involved in the treatment of home-care patients with COPD (Table 1). In the self-assessment, the application shows the patient’s own data as a radar-chart or a time-series graph on the screen, and data for the six items are displayed on the screen along with the extent of symptoms. The results of an overall evaluation of the six items are presented as the ‘Total Health Index’, which is unique to this application, and was calculated using an original formula. The ratings for each of the six assessment items (from 0 to 5) were added and multiplied by 20, and then divided by the number of items, i.e., six, to obtain the value of the Total Health Index (Equation (1)). As the target users of the system were elderly patients with COPD, we designed the operation method to be as easy as possible, with a minimal amount of information displayed.

We also built a network server for the tablet application where the patients’ data could be saved and accessed by support staff, such as doctors, nurses and nutritionists. In this study, we obtained the consent of the patients after explaining that only the three researchers in this study would operate the server and access the data stored in the server. Additionally, the researchers could share the patients’ personal information between themselves and their doctors. Finally, we developed a tablet application that enabled doctors and other support staff to check and assess data sent remotely from each patient’s tablet. These data included all items in the questionnaire regarding the extent of symptoms, as shown in Table 1. The patients were asked to complete the questionnaire every day, to provide a continuous record of their symptoms and thus enable rapid detection of any worsening of their health condition. Since the data could be shared between the doctor and patient, the doctor could select a suitable treatment method that matched the condition of the patient. The operating system used to develop the tablet application, which we named “Health Monitoring & Support Application”, was iPad OS 14 Xcode 13.0, and the Swift 5.5 programming language was used for software development. The application has the following functions: (1) monitoring of health conditions, (2) enabling early response to acute exacerbations; and (3) providing health education. The complete system uses two applications (one for patients and another for support staff, such as doctors, nurses and nutritionists).
(1)$$\begin{array}{l} x;\;The\;level\;of\;the\;six\;assessment\;items\;(0\;to\;5)\\ m;\;Total\;health\;index\\ m=\frac{20(x_{1}+x_{2}+\ldots +x_{n})}{n} \end{array}$$

### 3.1. Input & Send Health Condition and Physical Activity Data ‘APP for Patients’

#### 3.1.1. Input & Send Data Functions

Physical condition: extent of symptoms (cough, phlegm, respiration, sleep, appetite, vitality) assessed on a 5-level scale.Self-check that medications have been taken.Data that are inputted: number of steps per day and energy expenditure per day (kcal/day) measured using a Lifecorder (LC) pedometer (Suzuken, Nagoya, Japan).Tap on the button to transmit all daily data to the data server (Figure 2).

#### 3.1.2. Data Visualization Function for Patients

Users tap on a date in the calendar to display goal attainment levels in the chart on the right.A date is selected on the calendar.Numerical data for that date are displayed on a radar chart (Figure 3).

### 3.2. Remote Check and Assessment of Patient’s Health Condition ‘APP for Support Staff’ Enter the Patient’s ID Number

Select the numerical data item(s) for display.Specify the period of data to be displayed. Data are displayed as a time-series graph in the center of the screen.Mean data for the selected period are shown on a radar chart in the upper right corner (Figure 4).

## 4. Methods of Experimental Observation and Practical Assessment of the System

This study was conducted with the approval of the University of Toyama, National Institute of Technology, Toyama College, and National Hospital Organization Higashinagano Hospital.

The two participants in the study were a 74-year-old male with COPD (patient A) and a 65-year-old male with COPD (patient B). After an explanation of the study, the patients provided written consent to take part in the clinical evaluation for 8 months, from January 2021 to August 2021, in collaboration with two hospitals in Nagano Prefecture, Japan. Although patient A regularly visited the hospital for HOT, his condition was stable and did not interfere with his daily life. The condition of patient B was also stable when performing his daily life activities, and he did not require HOT. Table 2 and Table 3 list patient data on the following physical characteristics: sex, age, height, weight, body mass index (BMI), results of the 6-min walking distance test, pulmonary function, and disease stage.

The patients were required to use two devices in this study, a pedometer for daily monitoring of their step count and PA level, and an iPad with the monitoring system APP installed, for evaluation of their data and for sharing of their data with the support staff and their medical teams. The LC pedometer (Suzuken, Nagoya, Japan) is a waist-fitted pedometer (Figure 5). The built-in accelerometer of the LC classifies the intensity of physical activity into 11 stages (0–9 and micromovement) using a unique algorithm based on the data acquired from the vertical vibration and its frequency that occurs with motion of the body. The data are recorded in the LC every 4 s. The 11 PA intensity levels according to the LC were classified as light (PA levels 1–3), moderate (PA levels 4–6), and vigorous (PA levels 7–9). PA levels of 0 and micromovement indicate inactivity. According to the system of Kumahara et al., the energy cost of PA is expressed physiologically in terms of metabolic equivalents (METs) as a ratio of the metabolic rate, in which light-intensity activity is defined as <3 METs, moderate-intensity activity as 3–6 METs, and vigorous-intensity activity is >6 METs [14]. In the present study, PA was calculated as the total duration of PA per day at light, moderate, and vigorous intensity levels. Based on the age, sex, height and weight set in the LC, the number of steps and duration of PA by intensity (light, moderate, vigorous), walking distance and energy expenditure were calculated on a daily basis. LC has been used as the PA measurement sensor in many health-related studies. In addition, since patients are least resistant to wearing an LC compared to other sensors, the LC was adopted as the physical activity measurement sensor in this study [15]. The iPad used in the current study was an iPad Air (Apple, Cupertino, CA, USA) with cellular capability, to overcome any limitations of the Wi-Fi network environment at the patients’ houses. To ensure that the tablet was user-friendly, the number of icons on the display was kept to a bare minimum.

Patients were asked to wear the LC every day to measure their amount of daily PA for 8 months, from January to August 2021. The study was divided into the following three periods: a 2 month pre-remote monitoring and support period (pre-support period) from January to February 2021; a remote monitoring and support period (support period) from March to April 2021; and a further 4 month implementation period (from May-August 2021). In the first 2 month pre-support period (January to February 2021), we only measured the patients’ PA using the LC pedometer. During the second 2 month period, the support period (March to April 2021), in addition to continuing monitoring of their PA using the LC pedometer, the patients were asked to input their PA data recorded on the pedometer and their health data on the iPad APP, and to transmit the data to the support team. In the third period, the implementation period, from May to August 2021, the patients continued using the LC pedometer and iPad APP, as in the support period, to evaluate their utilization rate of our system over the 4 month implementation period.

Based on the data transmitted to the support team, the number of steps per day and the duration of PA per day (total duration of light-, moderate- and vigorous-intensity activity) were estimated in the pre-support and support periods, and the data from the two periods were compared to evaluate changes in the PA behavior of each participant due to use of the system. The changes in patient PA behavior between the first and second periods were also used to evaluate the utility of our system.

Furthermore, we held face-to-face meetings with each participant at the two participating hospitals in Nagano Prefecture and downloaded their detailed LC data to estimate their PA at a monthly medical examination for 8 months. We also met with their family doctors, who were provided with monthly reports of their health status. Additionally, we conducted interviews with the patients once a month and provided lifestyle guidance based on the data stored in the system for the period from March to August 2021.

A parametric test was utilised to compare the amounts of PA, using SPSS software (SPSS 16.0 Family; IBM Co., Chicago, IL, USA). We first performed a Levene test and confirmed that the two patients were homoscedastic. Next, we conducted a *t*-test with the significance level set to less than 5%.

## 5. Results

In patient A, the average number of steps per day and duration of PA per day were significantly higher during the support period than the pre-support period (1954 steps (591) vs. 1243 steps (680); 24.2 min (7.4) vs. 15.2 min (8.9), respectively; both *p* < 0.001) (Table 4, Figure 6). In patient B, although the average number of steps per day and total duration of PA per day improved after using the system, the increase was not significant (3253 steps (640) vs. 3474 steps (1003) steps; 39.7 min (8.1) vs. 42.9 min (12.9)).

The change in the amount of physical activity following use of the application, as well as the outcomes of guidance for living from meetings held at the two hospitals are presented below.

### 5.1. Patient A

Figure 7 shows changes in the levels of selected health indices per week in patient A from May to August 2021. The graph shows that patient A performed less PA when symptoms such as coughing and breathlessness were severe, and when he had insufficient sleep. PA improved along with improvements in his physical condition. In the hospital interviews with patient A, he said that he tried to limit his activity when he was not feeling well; therefore, we considered that he had appropriate self-management ability in relation to coexistence with COPD. In addition, collection of patient A’s previous PA data that had been measured over the course of his treatment for two years showed that the number of steps tended to decrease in the coldest part of winter and in the hottest part of summer. This kind of seasonal data will enable doctors and other support staff to consider personalized support methods, in anticipation of worsening COPD symptoms at these times.

Changes in the advice to patient A were as follows:⮚Pre-support period—Lifestyle guidance based on evaluation of only PA data

“Your PA duration is a little low, so do your best to walk and move in daily life”

⮚Support period—Lifestyle guidance using health monitoring system and PA data

“Your daily step count has decreased, are you feeling sick?”

“Do not overdo it as you may have coughing and other symptoms at the turn of the season.”

Before using the system, we asked patient A to increase his PA as part of exercise rehabilitation to prevent a decrease in exercise tolerability. The system enabled us to comprehensively monitor the patient’s PA and physical condition data and provide individualized lifestyle guidance. In this way, it might be possible to prevent acute exacerbations of COPD, and enable the early detection and treatment of acute exacerbations.

### 5.2. Patient B

Figure 8 shows changes in levels of selected health indices per week in patient B. The graph shows that patient B’s health index decreased on days of high PA levels, and that his health index recovered when his PA level was lower. Patient B performs regular work for a welfare bus transfer service. At that time, depending on the type of work, such as supporting people to get in and out of wheelchairs, his intensity of PA might increase. He is usually mindful to spend his time at home slowly, but when it comes to work, he sometimes feels unwell due to moderate or vigorous intensity PA, as is evident in Figure 8. These data led us to conduct interviews with patient B in which we discussed how he could successfully balance work with management of his physical condition.

Changes in the advice to patient B were as follows:⮚Pre-support period—Lifestyle guidance based on evaluation of only PA data

“You have been walking a lot, so it’s going well.”

“Try and maintain this condition.”

⮚Support period—Lifestyle guidance using health monitoring system and PA data

“Have you been overdoing it at work lately?”

“Please slow down and take it easy if you feel pain or have difficulty breathing.”

Before using the system, we had praised Patient B for increasing his PA but we had not noticed that the increased activity caused him knee pain. Using the system enabled us to advise him about developing the ability of self-management to control his PA level, in order to avoid worsening of his condition.

## 6. Discussion

The three pillars of nutrition, exercise and public participation are important in frailty prevention. Since frailty is highly prevalent in COPD patients, its prevention is an important issue for maintaining their physical, psychological and social QOL.

We developed a remote health support system for home-care patients with COPD that contributes to frailty prevention and tested its applicability and utility in two patients. In developing the application, we kept in mind that the users are elderly. Therefore, we made operation of the application as easy as possible, and deliberately presented only the essential information on the display. Previous researchers have developed a remote health monitoring system with multiple functions, but their system was hindered by the complexity of operation for elderly patients and their support teams [17]. In fact, many elderly patients refused to participate in the present study due to concerns about the operability of the system, which highlights the extreme importance of simplifying the system operation. In the implementation experiment, the two participants utilized the application for 4 months, from May to August 2021, at rates of 94.0% (110/117 days, patient A) and 94.9% (111/117 days, patient B). We consider that the convenience of the system enhanced its effectiveness in inducing a change in awareness in these two elderly patients with COPD, and that it could thus contribute to long-term frailty prevention. Other advantages of the system are that it allows remote management of the physical condition and PA of home-care patients, and a quick response to changes in medical conditions such as acute exacerbations. In addition, we attempted to incorporate patient education into the system, to foster patients’ self-management by taking an interest in their own physical condition. Accordingly, we tried to realize “data visualization” for the patient by taking advantage of the features of the tablet computer (iPad), and aimed to improve the patients’ own health management ability via this function.

In the implementation experiment using the developed system, one of the two COPD patients had a statistically significant increase in PA after using the application. The other patient had no statistically significant difference in PA after using the application, although his step count increased by 6.8% and his duration of PA increased by 8.1%. These data suggest that the patients modified their behavior for health promotion. Since step count is related to QOL and frailty [18], this behavior modification is considered to contribute, not only to preventing deterioration of the patient’s medical condition, but also to retention and improvement of QOL and prevention of frailty. Additionally, our developed system uses a pedometer to measure the amount of PA, since elderly patients are less resistant to using such devices. However, the system requires users to input their PA into the tablet manually, which could lead to input errors or forgetting to input the data.

A previous study reported that the rate of home exercise therapy in elderly patients with COPD is very low. Even understanding the need for exercise or PA is not always sufficient motivation for patients to increase their activity level [19]. Supervised high-intensity exercise therapy during hospitalization has a certain training effect, and while patients exercising on their own after discharge could have benefits, it is also associated with certain risks. On the other hand, it has been reported that, after discharge, low-intensity exercise, such as walking, is easier for elderly patients with COPD to continue at home [8]. Our system does not force patients to perform exercise, but expects patients to develop a self-management ability by recording data about their physical condition and PA in the application. From this perspective, the study results suggest the possibility of a beneficial effect, even though the number of subjects was small. Moreover, remote monitoring of the patients’ daily physical condition and PA enabled us to tailor lifestyle guidance to their own lifestyles and medical conditions. As the symptoms of COPD vary among individuals, the system shows future promise for providing personalized support in telemedicine. We believe that our developed remote health monitoring system will be useful in the medical and welfare fields.

In the implementation experiment, we approached remote health support from the following three perspectives: (1) prevention of physical frailty; (2) early response to acute exacerbations; and (3) health education for patients. It is difficult to obtain quantitative results for health support, including respiratory rehabilitation, for home-care elderly COPD patients. However, by connecting home-care elderly patients and their support team via ICT equipment, we found that it was possible to provide support in terms of physical as well as psychological and social aspects. In the future, we will continue to assist patients with COPD and their support teams by further developing the system to enable continuous support over a longer period of time.

## 7. Conclusions

An ICT-based support system, developed to provide comprehensive health support to elderly patients with COPD, was evaluated in a clinical setting. The amount of PA was significantly increased in one of the patients that used the system. Although the second patient showed no significant improvement in his PA levels, he did show a tendency for increased PA. Health and PA data submitted by the two patients with COPD enabled the provision of comprehensive and personalized health support for their individual lifestyles and medical conditions. In future development of the application, we will consider other devices that could make the system even more convenient to use. We will also add functions that support the aspect of social participation, which is important in frailty prevention. Full development of the system to encompass PA, nutrition and social participation will enable appropriate support to be provided to patients with COPD, and realization of its wider application in a variety of areas, including the medical and welfare fields.

## Figures and Tables

**Figure 1 sensors-22-02670-f001:**
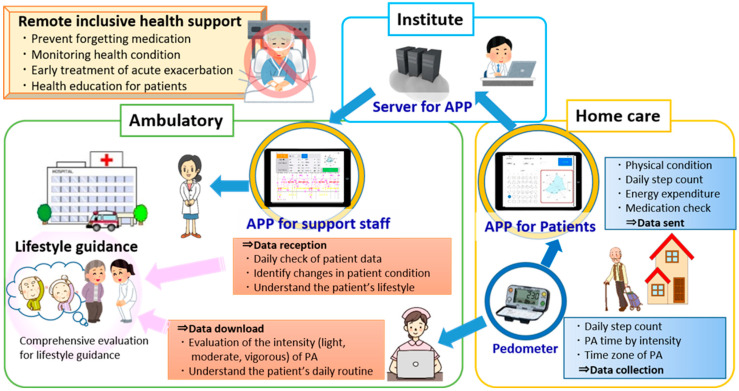
Outline of the developed remote health monitoring and support system.

**Figure 2 sensors-22-02670-f002:**
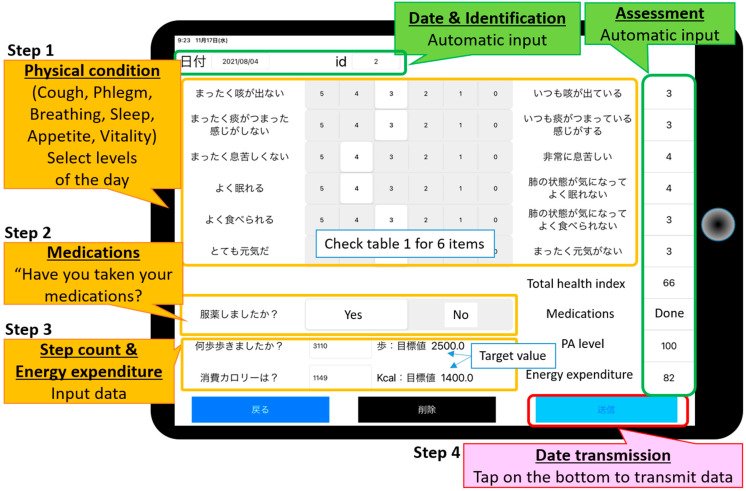
Data input and send screen of the tablet (APP for patients). The data inputted include (1) body condition; (2) medication check; (3) number of steps/day and daily energy expenditure. When the patient taps the “send” button, all data for that day are transmitted to our server.

**Figure 3 sensors-22-02670-f003:**
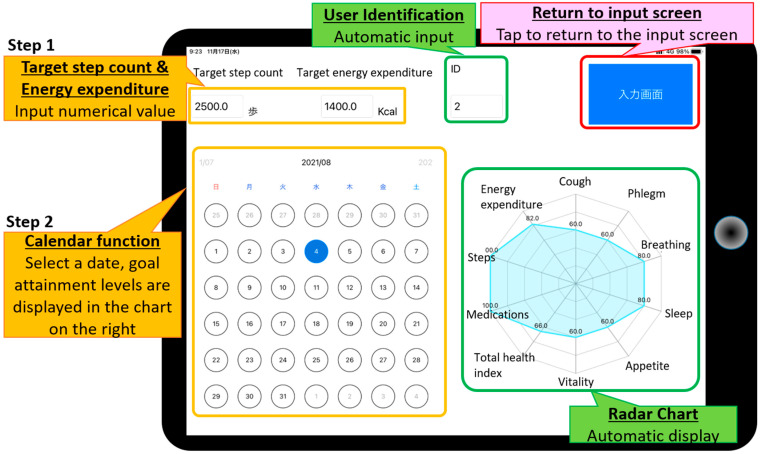
Patient application. Goal attainment levels are presented on a radar chart for the day selected (number of steps per day, duration of PA per day, numerical value of body condition). Patients can check their own data anywhere and anytime on the tablet device.

**Figure 4 sensors-22-02670-f004:**
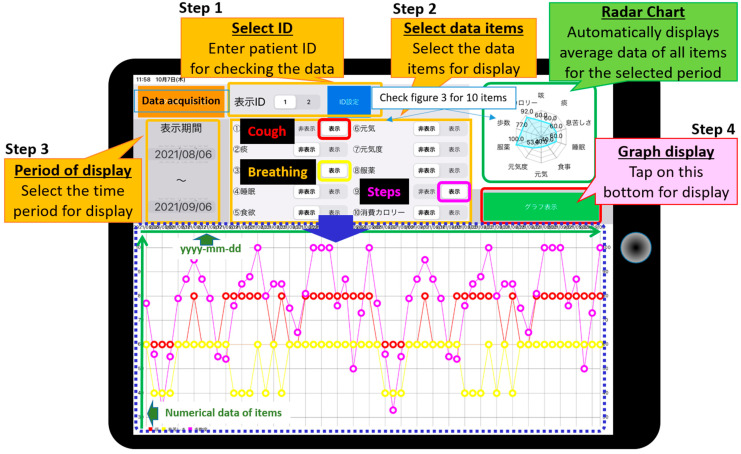
Example screen of the support staff application. A time series graph is displayed on the tablet device. The user selects the period and data items for display.

**Figure 5 sensors-22-02670-f005:**
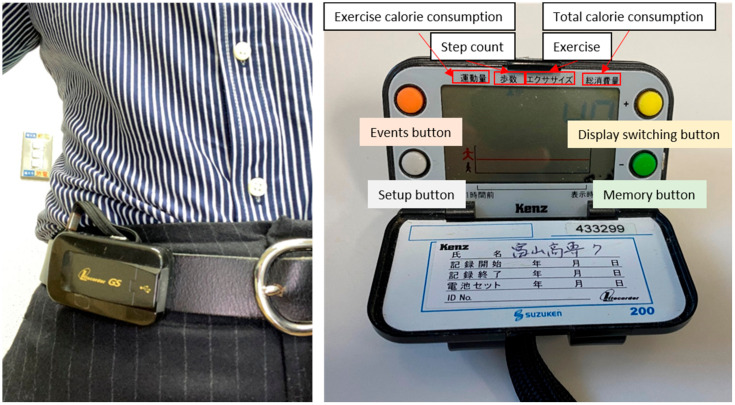
Images of the Lifecorder (LC). The image on the left shows how the LC is worn on the waist, and the image on the right shows the display of the LC.

**Figure 6 sensors-22-02670-f006:**
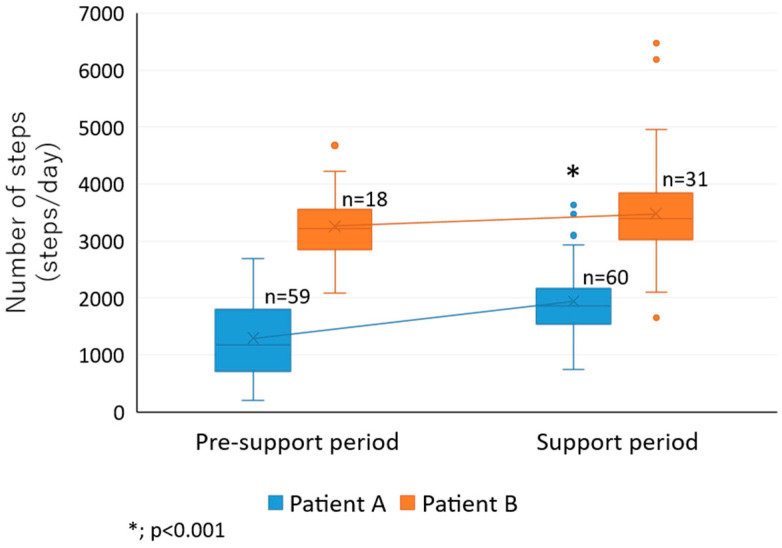
Box-and-whisker plots of the daily step counts in the pre-support and support periods in the two patients.

**Figure 7 sensors-22-02670-f007:**
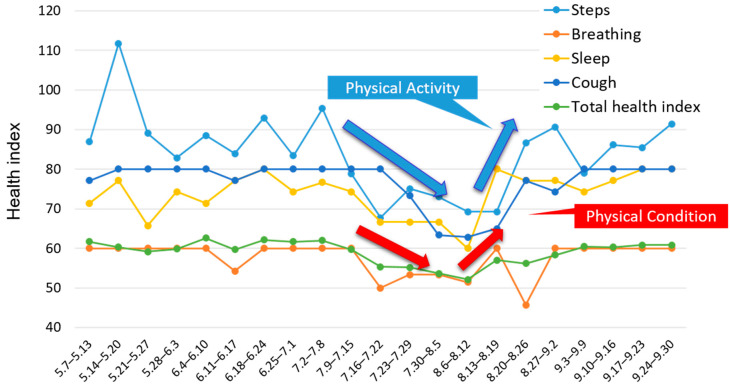
Average steps per week and levels of selected health items per week in patient A.

**Figure 8 sensors-22-02670-f008:**
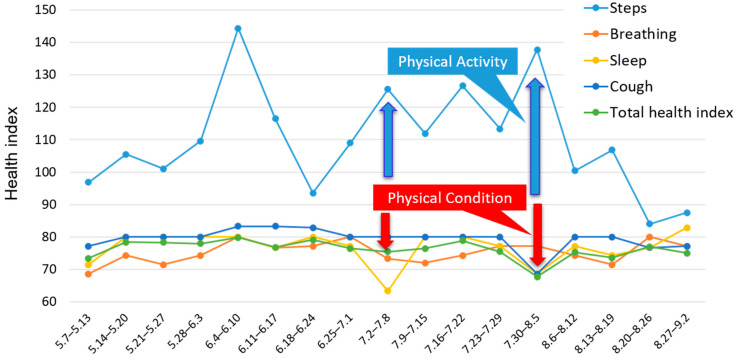
Average steps per week and levels of selected health items per week in patient B.

**Table 1 sensors-22-02670-t001:** Six assessment items in the COPD symptoms questionnaire.

Good Condition	Levels	Poor Condition
I don’t have a cough	5/4/3/2/1/0	I am always coughing
I don’t have any phlegm	5/4/3/2/1/0	I always have phlegm
I have no difficulty breathing	5/4/3/2/1/0	I have great difficulty breathing
I am sleeping well	5/4/3/2/1/0	I can’t sleep well because of my lungs
I am eating well	5/4/3/2/1/0	I can’t eat much because of my lungs
I feel very good	5/4/3/2/1/0	I do not feel well at all

**Table 2 sensors-22-02670-t002:** Participant characteristics.

Subject	Sex	Age (y)	Height (cm)	Weight (kg)	BMI	6 Minute Walking Distance ^1^ (m)	HOT ^2^
Patient A	Male	74	172.9	55.8	18.7	217	Yes
Patient B	Male	65	167.5	41.3	14.7	336	No

^1^ An exercise tolerability index. ^2^ HOT, home oxygen therapy.

**Table 3 sensors-22-02670-t003:** Pulmonary function and disease stage.

	Pulmonary Function ^1^					
Subject	FVC (L)	FEV1 (L)	FEV1% (%)	%FEV1 (%)	Stage ^2^	MRC Grade ^3^
Patient A	1.55	0.47	30.32	15.5	IV (Very severe)	4
Patient B	2.67	1.19	44.57	39.0	III (Severe)	3

^1^ FVC, forced vital capacity; FEV1, forced expiratory volume in one second; FEV1%, percent of forced expiratory volume in one second; %FEV1, percent predicted forced expiratory volume in one second. FVC is the total volume of air that can be exhaled with a maximum and slow exhalation. FEV is the volume of air that is exhaled with forceful expiration after maximum inhalation. Patients with COPD are unable to fully exhale, and their FEV is significantly lower than their FVC. The standard value of FEV is 3.00 to 4.00 L for men and 2.00 to 3.00 L for women. ^2^ According to the Global Initiative for Chronic Obstructive Lung Disease (GOLD), at Stage III, predicted FEV1 is 30–50%; Stage IV, predicted FEV1 is < 30% [2]. ^3^ Six-point MRC dyspnea scale, Japanese Respiratory Society (2003) [16]. Grade 3 dyspnea: walks slower than people of the same age on level ground or stops to catch breath while walking at own pace on level ground; Grade 4 dyspnea: stops for breath after walking approximately 100 yards or after a few minutes on level ground.

**Table 4 sensors-22-02670-t004:** Comparison of daily step count and duration of PA per day between the pre-support and support periods.

Subject	Number of Steps (Steps/Day) ^1^		Total Time of Physical Activity (Min/Day)	
	Pre-Support Period	Support Period	Pre-Support Period	Support Period
Patient A	1243 (680)	1954 (591) *	15.2 (8.9)	24.2 (7.4) *
Patient B	3253 (640)	3474 (1003)	39.7 (8.1)	42.9 (12.9)

Data are presented as the average (SD), *; *p* < 0.001, ^1^ Data for days on which the LC was worn for less than 8 h were excluded.

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
