# Peer review of "Development of a Remote Health Monitoring System to Prevent Frailty in Elderly Home-Care Patients with COPD"

_sensors, 2022, doi:10.3390/s22072670_

Round 1

Reviewer 1 Report

Dear Authors,

in your interesting manuscript, the following points should be added/changed to further improve it:

  • Generally, averages +- standard deviations need either brackets or doubled units, e.g. (40 +- 8) min in the abstract.
  • Lines 18-19: How can two strongly overlapping distributions be significantly different with p < 0.001? (ditto in section 5)
  • line 23: Please define ICT.
  • Please don't superscript the references.
  • Fig. 2: Please try making the letters in the English description labels a little bit larger, they are hard to read in the moment.
  • In Fig. 2, you mention "our server" - what about the patients' privacy, according to which rules may which people see the data? Is it necessary that patients sign an agreement that their doctors get an insight?
  • Fig. 4: Is there any description what the colors in the chart mean? And what is the x-axis - time in days?
  • Table 3: Please explain the medical terms a little bit more - whta is the forced vital capacity, what are good and bad values, etc.
  • line 219: Which test did you use, and how did you test the prerequisites?
  • Since the journal is called "Sensors", there should be more information about the sensors you use, incl. their reliability.

Author Response

I uploaded a file

Reviewer 2 Report

This study aimed to propose the development of a remote health monitoring system for home-care-elderly patients with COPD and evaluate its usability by two elderly home-care patients with COPD. I have the following major suggestions.

  1. What is the novelty of this study although several health monitoring systems have been proposed earlier? Please write down the contribution of the study at the end part of the Introduction section in bulleted form.
  2. The main drawback of this study is the small sample size (only two subjects with COPD). Authors should test their system with more number of participants.
  3. Authors should describe the state-of-art health monitoring system for numerous applications, such as mental workload, disease prediction, stress. Physiological signal-based health monitoring system was investigated for stroke prediction in a recent article, healthsos: Real-Time Health Monitoring System for Stroke Prognostics.
  4. Authors should improve reference by reporting studies related to the sensor-based health monitoring for disease prediction. For example, heart signal was investigated for stroke prediction in article, big-ecg: Cardiographic Predictive Cyber-Physical System for Stroke Management
  5. Section 1 and 2 should be combined. Introduction should consist of background study, purpose, and contribution. Authors should add more recent health monitoring studies, as suggested above, in introduction.
  6. Authors should mention the specification of pedometer. In addition, authors should include a sample real picture when a participant uses this system.
  7. Section 4 title should be Experimental methodology.
  8. Figure 1 is very naïve and simple. Authors should update this figure with more detailed information.
  9. Table caption must be given above the table. Please follow guidelines of journal manuscript preparation.
  10. Which statistical analysis did authors perform as a hypothesis test?
  11. How did the authors calculate health index? Authors should report algorithms and equations to define the health index as reported in results.
  12. Authors should include a few healthy subjects in this study to make a comparision of COPD patients and healthy adults.
  13. Results and discussion section need to be extended and improved. Authors must make discussion on the advantages and drawbacks of their proposed system with other studies adding a discussion section.
  14. From the writing point of view, the manuscript needs to be checked for typos and the grammatical issues should be improved.

Author Response

I uploaded a file.

Round 2

Reviewer 2 Report

Thanks for addressing comments. Still several suggestions were not addressed properly.

  • As suggested in comment 2 in the previous review, it was suggested to do an experiment with a greater number of people, which has not been addressed yet. experiment with Only two persons is not enough for statistical analysis and validation of the claim authors report in this study.
  • Still, the novelty and contribution of this study are not clear.
  • Written quality still needs to be improved. Authors should add scientific explanations of the basics of the technology they reported.
  • Manuscript needs to be checked for typos and the grammatical issues should be improved.
  • Authors should improve reference by reporting studies related to the sensor-based health monitoring systems as suggested in the last review.
  • Figure quality of results still needs to improve.
  • As suggested in comment 11 in the last review, the was authors write the equation is not appropriate. Authors should describe the parameters in detail. The equation should be written in text contents, not in Figure.
  • Authors need to compare their study with other studies in the discussion section in a tabular form.

Author Response

I am sorry for the delay in responding.
I upload a cover letter. Please check it again.
